# The effect of light conditions on the content of selected active ingredients in anatomical parts of the oyster mushroom (*Pleurotus ostreatus* L.)

**Agnieszka Zawadzka**[1�], **Anna Janczewska**[1�], **Joanna Kobus-Cisowska**[1�]*,
**Marcin Dziedziński**[1�], **Marek Siwulski**[2�], **Ewa Czarniecka-Skubina**[3�], **Kinga Stuper-Szablewska**[4�]

**1** Department of Gastronomy Science and Functional Foods, Poznan University of Life Sciences, Poznan, Poland, **2** Department of Vegetable Crops, Poznan University of Life Sciences, Poznan, Poland, **3** Department of Food Gastronomy and Food Hygiene, Warsaw University of Life Sciences, Warsaw, Poland, **4** Department of Chemistry, Poznan University of Life Sciences, Poznan, Poland

☉ These authors contributed equally to this work.
* joanna.kobus-cisowska@up.poznan.pl

**Data Availability Statement:** All relevant data are within the paper.

## Abstract

This study aimed to evaluate the effect of cultivation conditions in the context of light on the retention of selected vitamins, minerals and polyphenols in the stem and cap of the oyster mushroom *(Pleurotus ostreatus* L.*)*. Additionally, the effect of the retention of bioactive components on the antioxidant activity of mushroom extracts was evaluated, taking into account the morphological part. Oyster mushrooms grown in the light of 200 lux had higher riboflavin content compared to mushrooms exposed to the light of lower intensity. The thiamine content of the mushrooms dropped with decreasing light intensity during cultivation. The content of biologically active compounds was found to be equal in the stem and the cap. In the case of riboflavin, it was shown that its contents in cap fractions, irrespective of the cultivation method, was statistically significantly higher than in stems. The mineral composition of caps and stems differed from each other. No differences in Zn and Cu content between the morphological parts of the mushroom studied were found. However, it was shown that the stems, regardless of the type of light, contained less iron, magnesium and sodium. Thus, it was observed that limited light exposure caused an increase in the content of total polyphenolic compounds, which did not correlate with antioxidant activity. There was no effect of the light on the antioxidant activity of mushrooms. It was also shown that stem extracts had higher antioxidant activity compared to the extracts obtained from the caps. This findings point to the possibility and potentail of use both fraction of mushrooms in the new food products development.

## Introduction

Along with fruit and vegetables, edible mushrooms are now at the centre of interest of many research centres around the world. The interest is mostly focused on their flavour and aroma, but in fact, edible mushrooms have antifungal, anti-inflammatory, antiviral, antibacterial, hepatoprotective, antidiabetic, hypolipidemic, hypotensive and cytotoxic properties, as well as

**Funding:** The publication was co-financed within the agreement between Ministry of Education and Science (MEiN) and Poznan University of Life Sciences (UPP), grant number DWD/4/44/2020 from 23.11.2020r. The funders had no role in study design, data collection and analysis, decision to publish, or preparation of the manuscript.

**Competing interests:** The authors have declared that no competing interests exist.

a unique composition including such compounds as polyphenols, tocopherols, B vitamins, minerals and proteins [1–3]. Mushrooms are a rich source of highly and easily absorbable minerals. They have been proven to contain significant amounts of potassium, calcium and sodium. Trace amounts of such elements as iodine, fluorine, copper, zinc, mercury, and manganese are found in the mushroom fruiting bodies. Among edible mushrooms, oyster mushrooms contain significant amounts of B and D vitamins [4, 5]. Thiamine, riboflavin, niacin, pyridoxine, pantothenic acid, folic acid and provitamin D2 were found in their fruiting bodies. Fungal polysaccharides include chitin, α- and β-glucans, mannans, xylans, and galactans [6]. Polyphenols are mainly responsible for the antioxidant properties of oyster mushrooms.

Botanically, oyster mushrooms are classified under Basidiomycetes. The natural habitat of *Pleurotus* spp. is tropical and subtropical forests. In central European conditions, this species is found on dead trunks, logs and stumps of deciduous trees. It can also be grown on different substrates made from lignocellulosic waste [7]. The fruiting bodies of oyster mushrooms are often found in groups consisting of several larger and smaller specimens that grow out of a common base or are arranged in a tiled pattern one above the other. The fruiting bodies are a source of easily digestible protein, folic acid, amino acids, B vitamins and mineral salts. They have been found to contain lovastatin that lowers blood cholesterol levels. It also contains pleuran, the substance with immunostimulatory and anticancer effects [8].

In the cultivation of oyster mushrooms, the yield depends crucially on the climatic conditions. Oyster mushroom species and varieties require light of specific intensity to produce properly formed fruiting bodies. Light is not essential in the mycelial growth period. However, in the period of initiation and growth of fruiting bodies, it is a decisive factor for obtaining a high yield of good quality. The growth of fruiting bodies depends not only on the light intensity but also on the length of the light period in diurnal rhythm. The amount of light needed to develop fruiting bodies can be adjusted by decreasing the lighting duration while increasing light intensity at the same time. It is also possible to increase the lighting duration while decreasing light intensity [9]. Most importantly, however, light intensity was found to affect the morphological characteristics of the oyster mushroom, including cap size and stem length. An increase in the light intensity increases the cap size. As the stems are a waste product, the overarching goal of the current sustainable production is to achieve as high a proportion of the cap as possible compared to the stem. However, from the perspective of closed-loop functional food technology, it is advantageous to produce food also from residues, i.e. from the stem, so that every part of the cultivated raw material is utilised. Oyster mushrooms are often used for the production of functional food. It increases the nutritional value and modifies the taste of food [10]. On the other hand, studies indicate that the growing conditions discussed earlier have a decisive impact not only on the visual qualities of the mushrooms but also determine the content of those ingredients. The same is true for the anatomical part of the mushroom [11, 12]. The concentration of bioactive compounds can vary in the cap and stem and can be variable depending on the growing conditions.

This study aimed to evaluate the effect of cultivation conditions in the context of light on the retention of selected vitamins, minerals and polyphenols in the stem and cap of the oyster mushroom *(Pleurotus ostreatus* L*.)*. Additionally, the effect of the retention of bioactive components on the antioxidant activity of mushroom extracts was evaluated, taking into account the morphological part.

## Material

The substrate for the experiment was prepared by a professional company producing oyster mushroom substrate in Łobez near the city of Jarocin (Poland).

The substrate was prepared from wheat straw, cut into 4–5 cm long chaff and wheat bran. 5 kg of bran was added to 100 kg of dry straw. The mixture was moisturised with tap water to a 70% moisture level and pasteurised at 58–60˚C for 48 hours. Once the substrate was cooled to 25˚C, it was inoculated with the mycelium of the Spoppo strain of *Pleurotus ostreatus* (Sylvan Company).

The substrate was mixed with mycelium, which accounted for 3% in proportion to the wet weight of the substrate, and pressed into blocks using a specialised machine that placed the blocks in perforated film bags. Each substrate block weighed 15 kg. The incubation process was conducted at a temperature of 25˚C and relative air humidity of 85–90%. Once the substrate was completely covered with the mycelium, it was transferred to the cultivation chambers. Different conditions for the setting and growth of fruiting bodies were applied. The conditions variants were as follows: temperature 15±1˚C and light intensity of 200 lux for 8 hours a day until the appearance of 0.5–1 cm fruiting buds; afterwards, during the growth of fruiting bodies, three lighting variants of 200, 50 and 10 lux for 8 hours a day. The tested samples were coded according to the intensity of the light by assigning the corresponding codes: C200 is the fraction of the cap maturing at 200 lux, S200 marks of the stem maturing at 200 lux. C50 and S50 are the caps and stems of fruiting bodies maturing at 50 lux. The C10 and S10 cap and stem fractions matured under 10 lux lighting conditions (Fig 1). In the experiment, fluorescent lamps with colour temperature of 6500K (LF80, Philips, Netherlands) as source of the light.

## Methods

### Colour of the mushroom and extract

The colour of caps, stems and mushroom extracts was measured. Colour measurement was performed in $L^*$ $a^*$ $b^*$ CEN unit system using a CM-5 spectrometer (Konica Minolta, Japan) in accordance with the methodology described by the device producer. The light source used was D 65 and the colour temperature was 6504 K. The observation angle of the standard colourimetric observer was 10˚. Measurements for each sample were repeated fivefold. The instrument was calibrated using a black pattern.

### Analysis of thiamin and riboflavin content

The B1 and B2 vitamin content (thiamin, riboflavin) was analysed using the Acquity H UPLC system equipped with a Waters Acquity PDA detector (Waters, USA) after the prior enzymatic and acid extraction (Zdzieblo 2015). The enzymatic hydrolysis was carried out for 2 hours at 50˚C in the presence of tacadiastase, and acid hydrolysis was performed using 0.6 M hydrochloric acid in contact at 90˚C for one hour. An Acquity UPLC® BEH C18 column (50 mm × 2.1 mm, particle size 1.7 μm) (Waters, Ireland) was used for the determinations. The phase was formed by a precipitate of methanol and 0.05 M $NaH_2PO_4$ containing 0.005 M hexanesulfonic acid, pH 3.0. Gradient elution was performed. Thiamin was converted into thiochrome derivative with 1% potassium hexacyanoferrate (II) and immediately quantified with the Acquity column used. UPLC® BEH C18 column (150 mm × 2.1 mm, particle size 1.7 μm). The phase was formed by a precipitate of methanol and 0.05 M $NaH_2PO_4$ containing 0.005 M hexanesulfonic acid, pH 3.0. Gradient elution was performed. The flow rate was 0.4 ml/min. The concentrations of the analysed substances were determined using an internal standard at a wavelength of λ 267 nm for vitamins B1 and B2. Compounds were identified by comparing the retention time of the analysed peak with the retention time of the standard and by adding a specific amount of the standard to the analyzed samples and reanalysing them. The detection level was 1 μg/g.

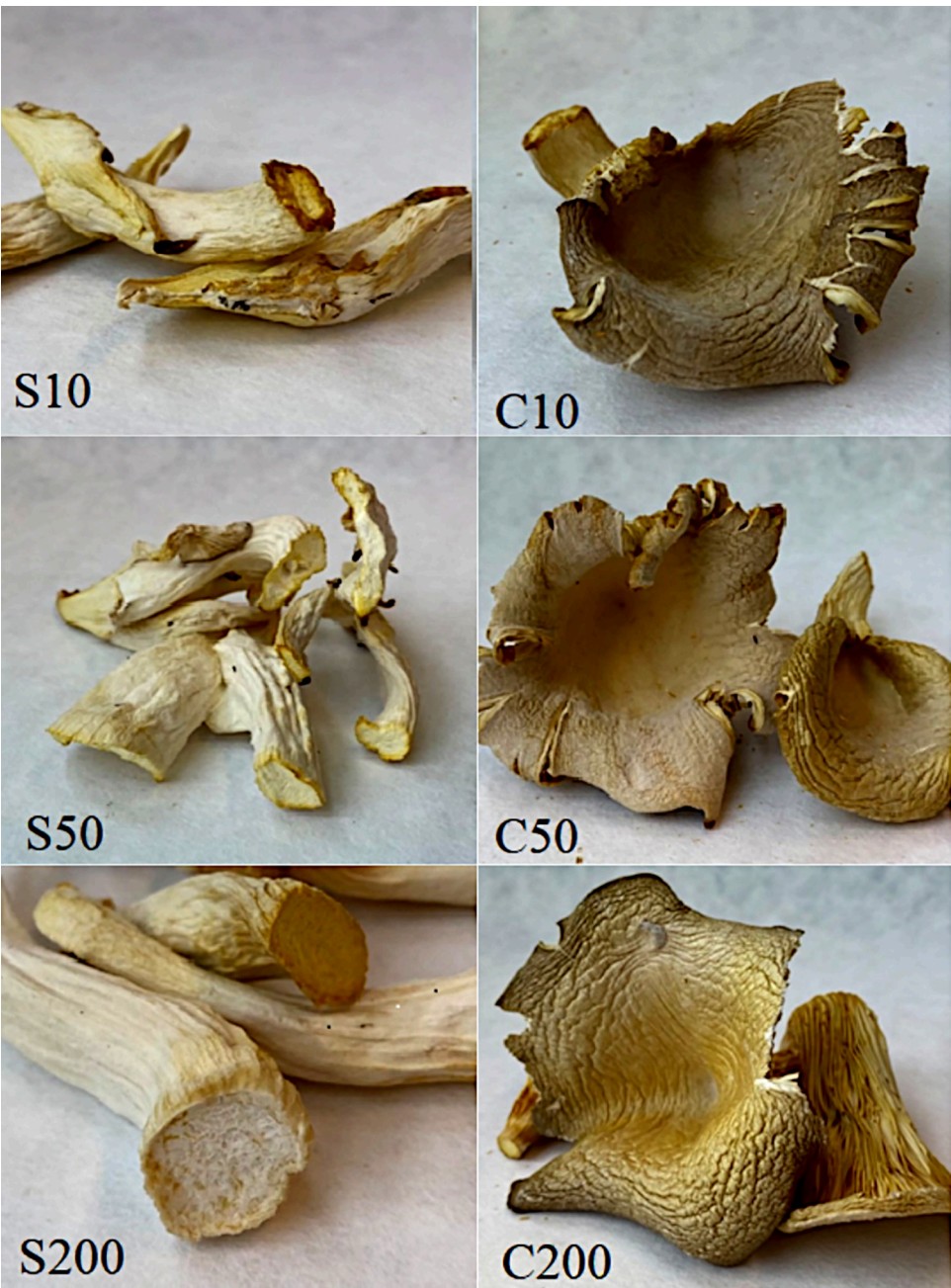

**Fig 1. Photo of the fruiting bodies of the oyster mushroom cultivated in the test conditions by anatomical part.**
Legend: C200 is the fraction of the cap maturing at 200 lux, S200 marks of the stem maturing at 200 lux. C50 and S50 are the caps and stems of fruiting bodies maturing at 50 lux. The C10 and S10 cap and stem fractions matured under 10 lux lighting conditions.

## Mineralisation and mineral content assay

Two aliquots of approximately 0.25 g were made from ground freeze-dried fruit, which were quantitatively transferred to Teflon vessels. 0.5 ml purified water, 5 ml 65% HNO3, 0.5 ml 37% HCl and 0.5 ml 5% $H_2O_2$ were added to each vessel. The prepared samples were subjected to microwave mineralisation in a closed system using a mineralisation programme divided into 5

successive stages with different parameters (temperature: 160˚C, 190˚C, 210˚C, 50˚C; pressure: 40bar; time: 5, 10, 15, 10, 1 min; power: 90, 99, 99, 0, 0%). After cooling, the mineralisates were quantitatively transferred to centrifuge tubes and diluted with purified water to 50 ml.

Control and measuring apparatus was used to determine the Fe, Cu, P, Zn, Mg and Na content; it includes the TOPwave microwave mineralisation system (Analytik Jena, Germany), an emission spectrometer with an inductively coupled plasma ICP OES model: PlasmaQuant PQ9000 (Analytik Jena, Germany), as well as an autosampler.

The system operation efficiency analysis was carried out by stabilising plasma in a flame and recalibrating the model solutions. Plasma power in ICP-OES analysis was 1200W, with a plasma gas flow rate of 12l/min, an auxiliary gas flow rate of 0.5l/min and atomising gas flow rate of 0.6l/min. The axial direction of measurement was set in the case of the Fe, Zn, Cu and P analysis while radial analysis was used for Mg and Na. The results were developed using the Spect PQ 1.2.3.0 software (Analitik Jena, Germany). The quantitative results of the analysis were automatically corrected for the average values for two blank samples, mineralised similarly to the real ones. All results are given in mg/100g of freeze-dried sample.

## Antioxidant activity

The extract's antiradical scavenging potential against DPPH radicals was also analysed. To that end, a methanolic solution of DPPH was used to evaluate the free-radical scavenging potential of the crude extract according to the method described before [13]. The degree of the solution's discolouration indicated the scavenging efficacy of the added substance. For this analysis, 1 ml of the extract solution was supplemented with 2 ml of pure methanol (Honeywell, United Kingdom), followed by 0.25 ml of 1 mM DPPH• ethanolic solution. The mixture was vortexed for ~60 s and left for 20 min at room temperature. Absorbance was recorded at $\lambda = 517$ nm (Meterek SP 830, Taiwan). Methanol was used to prepare a reference sample and the control. To plot a calibration curve, the absorbance values were measured simultaneously for samples containing respective concentrations of Trolox (Sigma-Aldrich, Germany) as a standard (0.5, 1.0, 1.5, and 2.0 mg/ml; $r^2 = 0.9639$). The results are expressed as % of inhibition.

The total polyphenol content was determined using Folin-Ciocalteu's reagent. Sample absorbance was measured at a wavelength of 765 nm with a UV-VIS spectrophotometer (Jena). Gallic acid was the standard and the results were expressed in mg/g dry weight of the extract according to Cheung et al. (2003) [14].

The metal chelating activity was measured as described by Kobus-Cisowska (2019), i.e. by adding 0.1 mM FeSO4 (0.2 mL) followed by 0.25 mM ferrozine (0.4 mL) into 0.2 ml of the extract [15]. Mixture absorbance after incubating at room temperature for 10 min was recorded at 562 nm. The chelating activity was calculated and described as %.

## Statistical analysis

Statistical analysis of all results was performed using Microsoft Excel 2013 software (USA) and Statistica 13 software (StatSoft, Poland). The $p$ values for Levene's test of independent variables were calculated.

## Results

The obtained extracts were examined for physicochemical properties, the results are shown in Table 1. The colours of the tested mushrooms and the extracts obtained from them were evaluated, which was shown to be statistically significantly different. In the case of the mushroom colour, the parameter $L^*$, responsible for determining the extract's lightness, reached the highest value in the C50 fraction at 73.99 and the lowest in S50 (59.04). The highest value of the

**Table 1. Characteristics of the studied anatomical parts of the oyster mushroom *(Pleurotus ostreatus* L.)* and their extracts expressed in units of L\*a\*b\*.**

| Sample name | Mushroom colour | | | Extract colour | | | RAL palette colour (extract) | RAL palette colour (mushroom) |
|---|---|---|---|---|---|---|---|---|
| | L* | a* | b* | L* | a* | b* | | |
| C200 | 67.68 [b] ± 1.19 | 6.36 [b] ± 0.66 | 16.93 [a] ± 0.55 | 44.81± 0.18 | 14.7 [b] ± 0.55 | 16.81± 0.41 | | |
| S200 | 64.00 [b] ± 2.15 | 6.93 [b] ± 0.87 | 27.77 [b] ± 1.19 | 42.28± 0.49 | 13.9 [a] ± 0.12 | 13.55± 0.09 | | |
| C50 | 73.99 [c] ± 2.22 | 4.81 [a] ± 0.81 | 26.67 [b] ± 0.76 | 42.54± 0.36 | 14.87 [b] ± 0.31 | 17.69± 0.05 | | |
| S50 | 59.04 [a] ± 1.89 | 6.42 [b] ± 1.54 | 18.49 [a] ± 0.65 | 44.12± 0.62 | 12.29 [a] ± 0.31 | 13.82± 0.10 | | |
| C10 | 68.46 [b] ± 1.44 | 7.19 [c] ± 0.87 | 23.13 [b] ± 0.13 | 41.72± 0.51 | 14.03 [b] ± 0.31 | 16.48± 0.11 | | |
| S10 | 70.78 [c] ± 0.78 | 5.58 [b] ± 1.23 | 17.18 [a] ± 0.65 | 43.46± 0.41 | 13.33 [a] ± 0.11 | 14.5± 0.03 | | |

The results are mean values of three determinations ± standard deviation. Values sharing the same letter in a column are not significantly different (P ≤ 0.05).

parameter a*, which is responsible for the colour change in the green to red range, occurred in the case of the S200 fraction and was 6.93, while the lowest value of C50 was 4.81. On the other hand, the parameter b*, responsible for the colour change from blue to yellow, took lower values for the fruiting bodies marked C200 (16.93) and higher values for that marked S200 (27.77). For the extracts tested, the results of the different fractions of a given parameter were at similar levels. The highest parameter L* value was achieved by sample C200 and amounted to 44.81 while the lowest value was achieved by sample C10 at 41.72. As far as the parameter a* of the extract is concerned, the highest and the lowest value among the samples tested were achieved by C50(14.87) and S50(12.29), respectively. The value of parameter b* is the highest for the C200(16.81) fraction and the lowest for the S200(13.55) fraction.

The thiamine and riboflavin content of the mushrooms analysed was tested, with the results shown in Table 2. The study showed that the S200 fraction had the highest content of thiamine (0.134) and the C10 sample had the lowest one (0.076). The riboflavin content in mushroom C200 (0.166) largely stood out as the highest among all fractions; on the other hand, sample

**Table 2. Vitamin content in extracts from the anatomical parts of the oyster mushroom (*Pleurotus ostreatus* L.).**

| Sample | Thiamine | Riboflavin |
|---|---|---|
| S200 | 0.134 [c] ±0.003 | 0.510 [b] ±0.003 |
| C200 | 0.125 [c] ±0.004 | 0.166 [a] ±0.095 |
| S50 | 0.130 [c] ±0.024 | 0.363 [b] ±0.030 |
| C50 | 0.107 [b] ±0.002 | 0.478 [b] ±0.023 |
| S10 | 0.126 [c] ±0.001 | 0.160 [a] ±0.002 |
| C10 | 0.076 [a] ±0.005 | 0.750 [c] ±0.018 |

The results are mean values of three determinations ± standard deviation. Values sharing the same letter in a column are not significantly different (P ≤ 0.05).

**Table 3. Mineral content in extracts from the anatomical parts of the oyster mushroom (*Pleurotus ostreatus* L.).**

| Mineral content (mg/100g) | variety | | | | | |
|---|---|---|---|---|---|---|
| | C200 | S200 | C50 | S50 | C10 | S10 |
| Fe | 1.15[c] ±0.01 | 1.01[b]±0.01 | 1.04[b]±0.01 | 0.96[a]±0.10 | 1.02[b]±0.10 | 0.98[a]±0.10 |
| Cu | 0.17[a] ±0.01 | 0.17[a]±0.03 | 0.21[b]±0.01 | 0.22[b]±0.01 | 0.20[b]±0.01 | 0.21[b]±0.01 |
| Zn | 0.65[c]±0.11 | 0.42[b]±0,15 | 0.63[c]±0.10 | 0.48[b]±0.10 | 0.49[b]±0.10 | 0.37[a]±0.10 |
| P | 143.22[b]±2.02 | 165.23[c]±4.43 | 155.21[c]±4.22 | 163.22[c]±2.18 | 134.51[a]±832 | 165.51[c]±3.21 |
| Mg | 16.26[b]±0.31 | 13.02[a]±0.12 | 17.12[b]±0.31 | 12.61[a]±0.12 | 15.54[b]±0.16 | 12.32[a]±1.86 |
| Na | 19.91[c]±0.03 | 13.01[a]±0.04 | 16.42[b]±0.02 | 13.98[a]±0.01 | 17.43[bc]±0.04 | 12.44[a]±0.06 |

The mean values in the line marked with different small letters indicate the significance of differences ($p \leq 0.05$).

S10 (0.160) had the lowest riboflavin content. Increased light exposure resulted in an increase in thiamine content in the fungal fractions tested.

The content of such minerals as iron, copper, zinc, potassium, magnesium and sodium in the analysed mushrooms was also investigated (Table 3). The sample with the 200 lux light supply had the highest iron content. The highest copper content was detected in the fungal fraction exposed to a 50 lux light supply. Samples of the mushroom cap grown at 200 lux and stem 10 showed the highest zinc and potassium content, respectively. Samples of caps exposed to 50 lux lighting were characterised by higher magnesium content; on the other hand, samples of caps exposed to 200 lux lighting had higher sodium content. The stem had higher thiamine content compared to the cap under all growing conditions. In terms of riboflavin content, higher values occurred in the cap. Increasing the light supply caused an increase in thiamine and riboflavin content in the fungal fractions tested, which was 76.32% and 218.75% higher, respectively, compared to the fractions exposed to the lowest amount of light.

The tested mushroom extracts were evaluated for antioxidant potential using spectroscopic methods (Table 4). It was found that light exposure as well as the anatomical parts of the fungi influenced the antioxidant potential of the extracts tested. The oyster mushroom extracts contained compounds that reacted with the Folin-Ciocalteu reagent. The highest content of such compounds was found in the extract from the mushroom cap exposed to 10 lux (18.65). This extract showed 140.3% higher activity compared to the extract made from mushroom stems obtained under the same light supply conditions. Additionally, the study results were made more thorough by determining the effect of the mushroom extracts using the DPPH radical assay. The extract obtained from the stem grown under 50 lux conditions was shown to scavenge radicals at 57.26; the antiradical activity for the extract from the stem grown under 200 lux illumination, as shown by the DPPH assay, was slightly lower at 55.04. The results of this testing were also confirmed by the chelating assays performed, which showed higher activity for the sample of the cap exposed to a 200 lux light supply, where the mean value for stem extracts was 36.77, which was 82.6% higher than that for cap extracts, with the latter amounting to 20.13.

**Table 4. Antioxidant activity of extracts from the anatomical parts of the oyster mushroom (*Pleurotus ostreatus* L.).**

| Sample / activity | C200 | S200 | C50 | S50 | C10 | S10 |
|---|---|---|---|---|---|---|
| TPC (mg GAE/g d.m.) | 6.18±0.66[ab] | 2.72±0.66[b] | 11.89±0.38[ab] | 4.53±0.70[ab] | 18.65±2.22[a] | 7.76±0.31[ab] |
| DPPH (% DPPH scavenging) | 21.69±7.24[ab] | 55.04±8.94[ab] | 22.22±3.00[ab] | 57.26±5.95[b] | 3.32±0.20[a] | 47.23±7.33[ab] |
| Metal chelating (% Fe++ metal chelating) | 29.38±5.39[a] | 27.29±8.25[ab] | 11.95±4.20[b] | 21.68±5.28[ab] | 19.08±5.79[ab] | 24.58±3.49[ab] |

The mean values in the line marked with different small letters indicate the significance of differences ($p \leq 0.05$).

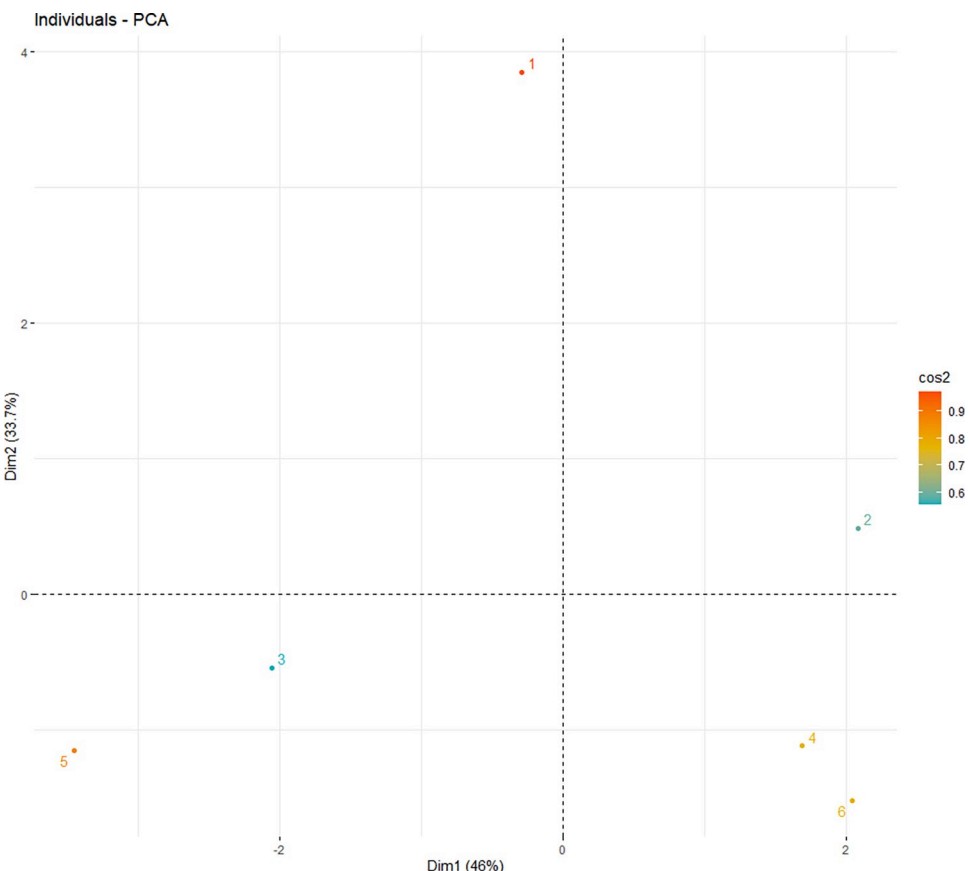

**Fig 2. Principal component analysis (PCA) biplot of individuals.** Legend: 1—C200; 2—S200; 3—C50; 4—S50; 5—C10; 6—S10; cos2 –square coordinates, quality of representation of the variables; Dim1 –principal component 1; Dim2 –principal component 2.

Graph of individuals (Fig 2). Individuals with a similar profile are grouped. A high cos2 indicates a good representation of the variable on the principal component. In this case, the variable is positioned close to the circumference of the correlation circle. A low cos2 indicates that the variable is not perfectly represented by the PCs. In this case, the variable is close to the centre of the circle.

Graph of variables (Fig 3) shows that positively correlated variables point to the same side of the plot. Negatively correlated variables point to opposite sides of the graph. Positively correlated variables are grouped together. Negatively correlated variables are positioned on opposite sides of the plot origin (opposed quadrants). The distance between variables and the origin measures the quality of the variables on the factor map. Variables that are away from the origin are well represented on the factor map.

## Discussion

Today, edible mushroom production is developing rapidly across the world. Cultivated mushrooms are valued for their relatively high vitamin, micronutrient and bioactive compound content. Many mushroom species can be classified as the so-called "functional foods", which are rapidly growing in importance in both food technology and nutrition [16]. Various oyster mushroom species occur naturally in a plethora of habitats and across numerous climate zones [17]. The following species are of the greatest importance in terms of production:

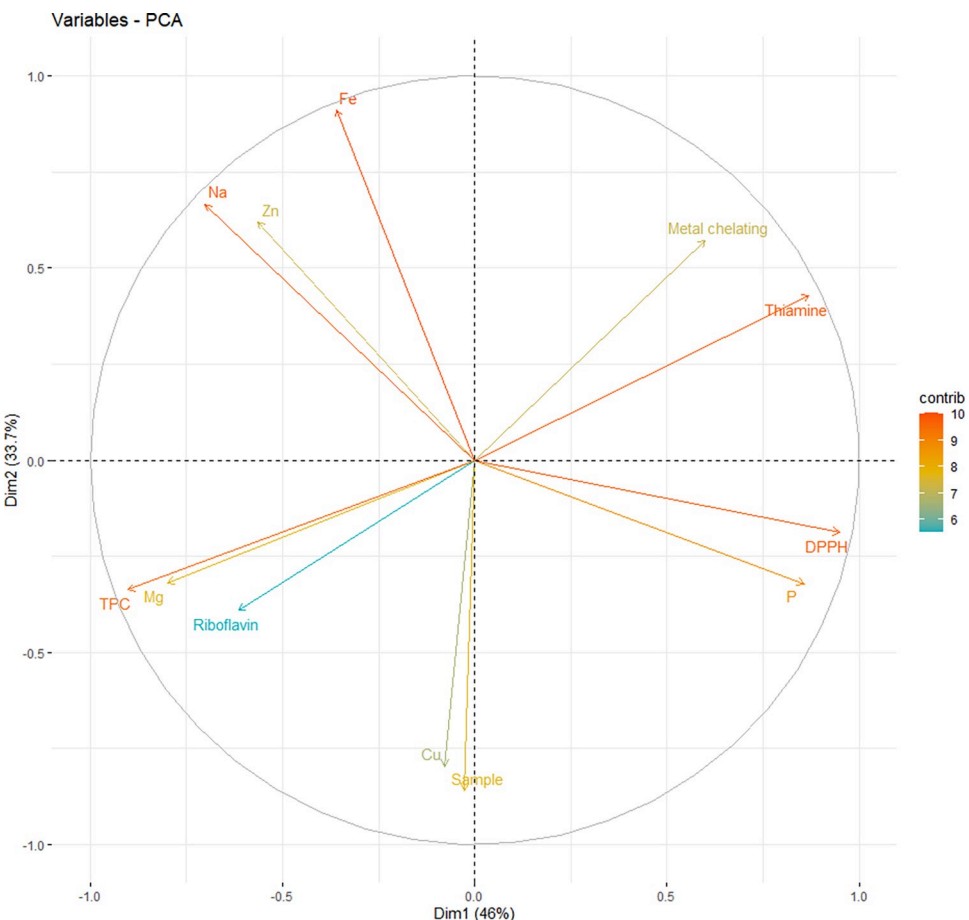

**Fig 3. Principal component analysis (PCA) plot of individuals.** Legend: Dim1 –principal component 1; Dim2 – principal component 2; contrib—contributions of the individuals to the principal components.

*Pleurotus ostreatus* (Fr.) Kumm., *P. pulmonarius* (Fr.) Quel., *P. columbinus* (Fr.) *Quel.*, *P. cornucopiae* (Pers.) Rolland, *P. eryngi* (Fr.) Quel., as well as *P. sajor-caju* [18]. However, irrespective of the fungus species, it is typically the caps that are of industrial importance. Caps are the trade commodity while stems are deemed production waste. On the other hand, the scientific approach to sustainable production aims to optimise growing conditions in such a way as to attain the highest possible cap-to-stem proportion. Numerous studies have confirmed that limiting light supply increases the cap's proportion. In this study, it was shown that the light supply provided to the mycelium tested had a major impact on the content of bioactive components in the oyster mushroom. Oyster mushrooms grown in the light of 200 lux had higher riboflavin content compared to mushrooms exposed to the light of lower intensity. In the other study on *Cordyceps militaris* researchers also showed that light intensity impacts the fruiting body yield bioactive compound production [19]. However, in mentioned studies light intensites of 1250, 1750 and 2500 lux were used, and 1750 lux was the most effective in case of fruiting body yield and biological efficiency [19]. In current study the thiamine content of the mushrooms dropped with decreasing light intensity during cultivation. At the same time, it was found that the biologically active compound content differed depending on the morphological part–it was different in the stem and in the cap. The mineral composition of caps and stems differed from each other. No differences in Zn and Cu content between the

morphological parts of the mushroom studied were found, however, it was shown that the stems, regardless of the type of light, contained less iron, magnesium and sodium. Limited light exposure caused an increase in the content of total polyphenolic compounds, which did not correlate with antioxidant activity. There was no effect of the light on the antioxidant activity of mushrooms, but it was shown that stem extracts had higher antioxidant activity compared to cap extracts. Other studies showed also that not only intensity, but also color of the light during development of fruit body can be important. Jang et al. showed that using blue and white LED irradiation makes it possible to obtain higher commercial yields of *P. ostreatus* compared to fluorescent light, and the ergothioneine was also the highest at the blue and white LED [20].

According to the literature it usually takes 5–8 days for the fruiting bodies of one crop to mature [21]. During the conducted observations of the effect of light on the development of vegetative and reproductive mycelium, it was found that the amount of fruiting bodies produced under exposure to light was over 100% higher compared to fully formed fruiting bodies growing with limited light. Changes in the number of fruiting bodies produced by mycelium with limited light supply showed differences in the dynamics of mycelium development. With ample light supply, cap-forming fruiting bodies are visible, whereas for no-light cultivation, only vegetative hyphae are noticeable [21]. However, the impact of these conditions on bioactive component retention was not determined.

It is generally considered that fungi are a better source of bioactive compounds compared to plants because not only can fungi form fruiting bodies faster, but their fruiting bodies can also be obtained using biotechnological methods, optimising the culture conditions accordingly to obtain the required amounts of active compounds [16, 22]. Different growing conditions affect not only vitamin and mineral content but also polyphenol content. The content of p-hydroxybenzoic acid was determined in *B. badius* (1.28 mg/kg d.w.), *B. edulis* (1.94 mg/kg d. w.), *C. cibarius* (2.30 mg/kg d.w.), with the highest content found in the species *P. ostreatus* (3.60 mg/kg d.w.). Synapic acid, whose content ranged from 2.11 to 14.29 mg/kg d.w., was also detected in A. *mellea, C. cibarius, L. deliciosus and P. ostreatus*. On the other hand, cinnamic acid content ranged from 1.09 to 4.06 mg/kg d.w. in *C. cibarius*, L. *deliciosus and P. ostreatus*, with the highest content found in B. badius (8.73 mg/kg d.w.) [23].

In this study, it was observed that limited light exposure caused an increase in the content of total polyphenolic compounds; however, this did not correlate with antioxidant activity. There was no effect of the light on the antioxidant activity of mushrooms, but it was shown that stem extracts had higher antioxidant activity compared to cap extracts. The flavonoid antioxidant activity depends on the conjugated double bonds at the C-2 and C-3 positions, as well as the hydroxyl groups and the carboxyl group at the C-4 position. The direct antioxidant action mechanism of these compounds consists in scavenging oxygen free radicals and reactive oxygen species and reduce their production in cells by inhibiting the activity of oxidative enzymes (e.g. lipoxygenase), and by readily donating hydrogen from the carboxyl group to reduce peroxides and hydroperoxides [24].

## Summary

The effect of light of the tested mycelium has a significant effect on the content of bioactive components in oyster mushrooms. Oyster mushrooms grown in the light of 200 lux had higher riboflavin content compared to mushrooms exposed to the light of lower intensity. The thiamine content of the mushrooms dropped with decreasing light intensity during cultivation. At the same time, it was found that the biologically active compound content differed depending on the morphological part–it varied in the stem and the cap. In the case of riboflavin, it was

shown that its contents in cap fractions, irrespective of the cultivation method taking into account the light intensity, was statistically significantly higher than in stems.

It was demonstrated that the mineral composition of caps and stems differed from each other. No differences in Zn and Cu content between the morphological parts of the mushroom studied were found, however, it was shown that the stems, regardless of the type of light, contained less iron, magnesium and sodium. Thus, it was observed that limited light exposure caused an increase in the content of total polyphenolic compounds, which did not correlate with antioxidant activity. There was no effect of the light on the antioxidant activity of mushrooms, but it was shown that stem extracts had higher antioxidant activity compared to cap extracts. Therefore, it is reasonable to develop a new direction for utilising oyster mushroom stems aimed at functional food production. Further work is needed to confirm the oyster mushroom stems' technological properties and how such properties may determine the qualities of new functional food products.

## Author Contributions

**Conceptualization:** Agnieszka Zawadzka, Joanna Kobus-Cisowska, Marek Siwulski, Ewa Czarniecka-Skubina.

**Data curation:** Joanna Kobus-Cisowska, Marcin Dziedziński, Marek Siwulski.

**Formal analysis:** Agnieszka Zawadzka, Anna Janczewska, Kinga Stuper-Szablewska.

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
