## [Decision Letter · Decision Letter 0]

26 Oct 2021

PONE-D-21-29872The effect of cultivation conditions on the content of selected active ingredients in anatomical parts of the oyster mushroom (Pleurotus ostreatus L.)PLOS ONE

Dear Dr. Dziedziński,

Thank you for submitting your manuscript to PLOS ONE. After careful consideration, we feel that it has merit but does not fully meet PLOS ONE’s publication criteria as it currently stands. Therefore, we invite you to submit a revised version of the manuscript that addresses the points raised during the review process.

The study "The effect of cultivation conditions on the content of selected active ingredients in anatomical parts of the oyster mushroom(Pleurotus ostreatus L.)" is an interesting study where specifically effect of light intensity and duration on contents of fruit bodies of oyster mushroom is studied. Kindly consider the points raised by reviewers and myself and revise the manuscript accordingly. 

1. As the study mainly focusses on the effect of light on active ingredients on different parts of the oyster mushroom, thus the title may be revised accordingly.

2. The language in the abstract is too complicated using multiple statements in a single sentence. Thus the syntax of the sentences must be taken care off.

3. Introduction part is too lengthy, may be reduced/limited to relevant statements with regard to the present study.

4. In all over the manuscript, long sentences are used, which may be revised and short and clear sentences may be used.

5. In introduction word variety is used in context of oyster mushroom, I am not sure whether it is variety of the same species or different species.

6. Careful Language correction is required in the manuscript.

We look forward to receiving your revised manuscript.

Kind regards,

Shwet Kamal, Ph.D

Academic Editor

PLOS ONE

Journal Requirements:

 “The publication was co-financed within the framework of the Ministry of Science and Higher Education program as “Regional Initiative Excellence” in years 2019–2022, project number 005/RID/2018/19 and by statutory funds of the Department of Gastronomy Sciences and Functional Foods of Poznan University of Life Sciences, grant number 506.751.03.00.”

5. Please ensure that you refer to Figure 1 in your text as, if accepted, production will need this reference to link the reader to the figure.

Additional Editor Comments:

The study "The effect of cultivation conditions on the content of selected active ingredients in anatomical parts of the oyster mushroom(Pleurotus ostreatus L.)" is an interesting study where specifically effect of light intensity and duration on contents of fruit bodies of oyster mushroom is studied. My observations are

1. As the study mainly focusses on the effect of light on active ingredients on different parts of the oyster mushroom, thus the title may be revised accordingly.

2. The language in the abstract is too complicated using multiple statements in a single sentence. Thus the syntax of the sentences must be taken care off.

3. Introduction part is too lengthy, may be reduced/limited to relevant statements with regard to the present study.

4. In all over the manuscript, long sentences are used, which may be revised and short and clear sentences may be used.

5. In introduction word variety is used in context of oyster mushroom, I am not sure whether it is variety of the same species or different species.

6. Careful Language correction is required in the manuscript.

Reviewers' comments:

Reviewer's Responses to Questions

**Comments to the Author**

1. Is the manuscript technically sound, and do the data support the conclusions?

Reviewer #1: Partly

Reviewer #2: Yes

2. Has the statistical analysis been performed appropriately and rigorously? 

Reviewer #1: Yes

Reviewer #2: Yes

3. Have the authors made all data underlying the findings in their manuscript fully available?

Reviewer #1: Yes

Reviewer #2: Yes

4. Is the manuscript presented in an intelligible fashion and written in standard English?

Reviewer #1: Yes

Reviewer #2: Yes

5. Review Comments to the Author

Reviewer #1: 1. Line no. 63, "Winter varieties require temperatures below 15ºC". Is it is winter varities or winter species?

2. Line no. 63 Check sign of degrees and correct.

3. Line no. 64 Check sign of degrees and correct.

4. Line no. 66 Check sign of degrees and correct.

5. Line 76-78 need refernce to support

6. Authors can specify that why some particular vitamins are taken in study, if any particular reason for that?

7. Improve discussion part

8. Mention the source and colour of light used in experiment

Reviewer #2: This manuscript tries to identify effect of cultivation conditions on the content of selected active ingredients in parts of the oyster mushroom that would help to develop nutritious and healthy mushroom production. This study is preliminary conventional because there have been larger trials that have shown similar findings. There are few typographical errors in the text, that must be corrected before the paper is accepted.

P1, L 21: Typographical errors

P4, L 92: Typographical errors

P9, L 207-208: Typographical errors

P10, L 220: Typographical errors

P11, L 237: Typographical errors

P12, L 239; Table 4: Typographical errors

P14, L 290: Typographical errors

P16, L 324, 325: Typographical errors

6. PLOS authors have the option to publish the peer review history of their article (what does this mean?). If published, this will include your full peer review and any attached files.

Reviewer #1: No

Reviewer #2: No

---

## [Author Response · Author response to Decision Letter 0]

9 Nov 2021

Dear Editiors and Reviewers,

Thank you for your thorough review. All comments and recommendations indicated by Reviewers have been included in the revised version of the manuscript. Changes to the manuscript were made in the content of the publication. The comments listed in the reviews are referenced in below.

Response to the editor:

1. As the study mainly focusses on the effect of light on active ingredients on different parts of the oyster mushroom, thus the title may be revised accordingly.

The title has been revised, and changed to: “The effect of light conditions on the content of selected active ingredients in anatomical parts of the oyster mushroom (Pleurotus ostreatus L.)”

2. The language in the abstract is too complicated using multiple statements in a single sentence. Thus the syntax of the sentences must be taken care off.

The syntax was improved and sentences shortened. 

3. Introduction part is too lengthy, may be reduced/limited to relevant statements with regard to the present study.

The introduction part has been shortened. 

4. In all over the manuscript, long sentences are used, which may be revised and short and clear sentences may be used.

5. In introduction word variety is used in context of oyster mushroom, I am not sure whether it is variety of the same species or different species.

It has been corrected or these parts has been deleted.

6. Careful Language correction is required in the manuscript.

Language correction across manuscript has been conducted.

Response to the Reviewer no. 1

Reviewer #1:

 1. Line no. 63, "Winter varieties require temperatures below 15ºC". Is it is winter varities or winter species?

2. Line no. 63 Check sign of degrees and correct.

3. Line no. 64 Check sign of degrees and correct.

4. Line no. 66 Check sign of degrees and correct.

This issues have been resolved. 

5. Line 76-78 need refernce to support

Reference has been added.

6. Authors can specify that why some particular vitamins are taken in study, if any particular reason for that?

Such vitamins are the most present in oyster mushrooms. 

7. Improve discussion part

Discussion has been improved.

8. Mention the source and colour of light used in experiment

This information has been added in the “material” section.

Response to the Reviewer no. 2

Reviewer #2: 

This manuscript tries to identify effect of cultivation conditions on the content of selected active ingredients in parts of the oyster mushroom that would help to develop nutritious and healthy mushroom production. This study is preliminary conventional because there have been larger trials that have shown similar findings. There are few typographical errors in the text, that must be corrected before the paper is accepted.

P1, L 21: Typographical errors

P4, L 92: Typographical errors

P9, L 207-208: Typographical errors

P10, L 220: Typographical errors

P11, L 237: Typographical errors

P12, L 239; Table 4: Typographical errors

P14, L 290: Typographical errors

P16, L 324, 325: Typographical errors

Typographical errors have been corrected.

We hope that the current state of the manuscript meets the standards of the Journal and that publication will be possible

Best regards,

Marcin Dziedziński

---

## [Decision Letter · Decision Letter 1]

21 Dec 2021

The effect of light conditions on the content of selected active ingredients in anatomical parts of the oyster mushroom (Pleurotus ostreatus L.)

PONE-D-21-29872R1

Dear Dr. Dziedziński,

The Manuscript entitled "The effect of light conditions on the content of selected active ingredients in anatomical parts of the oyster mushroom (Pleurotus ostreatus L.)" is an interesting piece of study. The study is well planned and inferred meticulously. The study describes changes in nutritional contents with regards to Vit B and D and other minerals with varying light intensities. Light play a crucial role in development of mushroom fruit bodies and also affects their quality. Researches have been conducted to study effect of light on fruiting and yield of Pleurotus mushroom but the study of nutritional contents in different parts of the fruitbodies due to changes in light intensity is a novel piece of work. 

We’re pleased to inform you that your manuscript has been judged scientifically suitable for publication and will be formally accepted for publication once it meets all outstanding technical requirements.

Kind regards,

Shwet Kamal, Ph.D

Academic Editor

PLOS ONE

Additional Editor Comments (optional):

The Manuscript entitled "The effect of light conditions on the content of selected active ingredients in anatomical parts of the oyster mushroom (Pleurotus ostreatus L.)" is an interesting piece of study. The study is well planned and inferred meticulously. The study describes changes in nutritional contents with regards to Vit B and D and other minerals with varying light intensities. Light play a crucial role in development of mushroom fruit bodies and also affects their quality. Researches have been conducted to study effect of light on fruiting and yield of Pleurotus mushroom but the study of nutritional contents in different parts of the fruitbodies due to changes in light intensity is a novel piece of work.

The Study can now be accepted for publication in PLOS ONE journal after taking the reviewer's comment.

Reviewers' comments:

Reviewer's Responses to Questions

**Comments to the Author**

1. If the authors have adequately addressed your comments raised in a previous round of review and you feel that this manuscript is now acceptable for publication, you may indicate that here to bypass the “Comments to the Author” section, enter your conflict of interest statement in the “Confidential to Editor” section, and submit your "Accept" recommendation.

Reviewer #1: All comments have been addressed

Reviewer #2: All comments have been addressed

2. Is the manuscript technically sound, and do the data support the conclusions?

Reviewer #1: Yes

Reviewer #2: Partly

3. Has the statistical analysis been performed appropriately and rigorously? 

Reviewer #1: Yes

Reviewer #2: Yes

4. Have the authors made all data underlying the findings in their manuscript fully available?

Reviewer #1: Yes

Reviewer #2: Yes

5. Is the manuscript presented in an intelligible fashion and written in standard English?

Reviewer #1: Yes

Reviewer #2: Yes

6. Review Comments to the Author

Reviewer #1: (No Response)

Reviewer #2: The majority of suggested changes have been made to this revised manuscript. There are still a lot of typographical errors. Please fix it.

7. PLOS authors have the option to publish the peer review history of their article (what does this mean?). If published, this will include your full peer review and any attached files.

Reviewer #1: No

Reviewer #2: No

---

## [Editor Report · Acceptance letter]

23 Dec 2021

PONE-D-21-29872R1 

The effect of light conditions on the content of selected active ingredients in anatomical parts of the oyster mushroom *(Pleurotus ostreatus L.)*

Dear Dr. Dziedziński:

I'm pleased to inform you that your manuscript has been deemed suitable for publication in PLOS ONE. Congratulations! Your manuscript is now with our production department. 

Kind regards, 

on behalf of

Dr. Shwet Kamal 

Academic Editor

PLOS ONE